# Natural Analogs to Ocean Alkalinity Enhancement

Adam V. Subhas[1], Nadine Lehmann[2], Rosalind E.M. Rickaby[3]

[1]Department of Marine Chemistry and Geochemistry, Woods Hole Oceanographic Institution, Woods Hole, MA USA.

[2]Institute for Marine and Antarctic Studies, University of Tasmania, Tasmania, AUS.

[3]Department of Earth Sciences, University of Oxford, Oxford, UK.

*Correspondence to*: Adam V. Subhas (asubhas@whoi.edu)

**Abstract.** Ocean alkalinity enhancement (OAE) research can be supplemented by studying the natural alkalinity cycle. In this chapter, we introduce the concept of natural analogs to ocean alkalinity enhancement. We describe Earth system processes relevant to OAE deployment and its measurement, reporting, and verification. We then describe some suitable natural analog locations that could serve as study sites to understand how these processes may interact with OAE. Approaches to examining the geological record are also considered. Practical considerations for establishing a natural analog study are discussed, including geochemical mass balance; choosing a site; establishing a control; choosing a measurement suite and platform; and coordinating with ocean models. We identify rivers and their plumes, glacial fjords, whiting events, and basinal seas with elevated alkalinity, as promising candidates for initial natural analog studies. This chapter is not meant to be prescriptive, but instead is written to inspire researchers to creatively explore the power of natural analogs to advance our understanding of OAE. Key recommendations include considering appropriate spatial and temporal scales of the study and associated measurement criteria, and designing the study with applicable outcomes to OAE research, including implications for deployment and/or monitoring.

## 1.1 Alkalinity cycling and a definition of natural analogs for OAE

Despite its residence time of about 100,000 years, there is a vigorous and dynamic alkalinity cycle in the ocean. The spatial and temporal patterns of alkalinity concentrations and fluxes are intimately linked with ocean biogeochemistry. Organic carbon production and remineralization cycles alkalinity through redox processing of oxygen and other electron acceptors (Froelich et al., 1979). Calcium carbonate ($CaCO_3$) production and dissolution consumes and produces alkalinity from the reef (Broecker and Takahashi, 1966, Andersson, 2015) to the ocean-basin scale (Emerson et al., 2011, Feely et al., 2002). The inventory of biogenic $CaCO_3$ accumulated in deep ocean sediments has long been recognized as a source of alkalinity over glacial-interglacial timescales, and will likely neutralize a significant fraction of fossil fuel-derived carbon dioxide ($CO_2$, Archer et al., 1998). Thus, the alkalinity cycle exerts its own unique influence – through multiple processes and scales – on the ocean's capacity to take up and store atmospheric $CO_2$.

Many OAE approaches are based on established geochemical weathering and acid-base reactions, and deploying these approaches will benefit from an understanding of earth's natural processing of alkalinity. These processes operate all around us, right now, at climate-relevant scales. The chemical and physical weathering of terrestrial rocks produces alkalinity

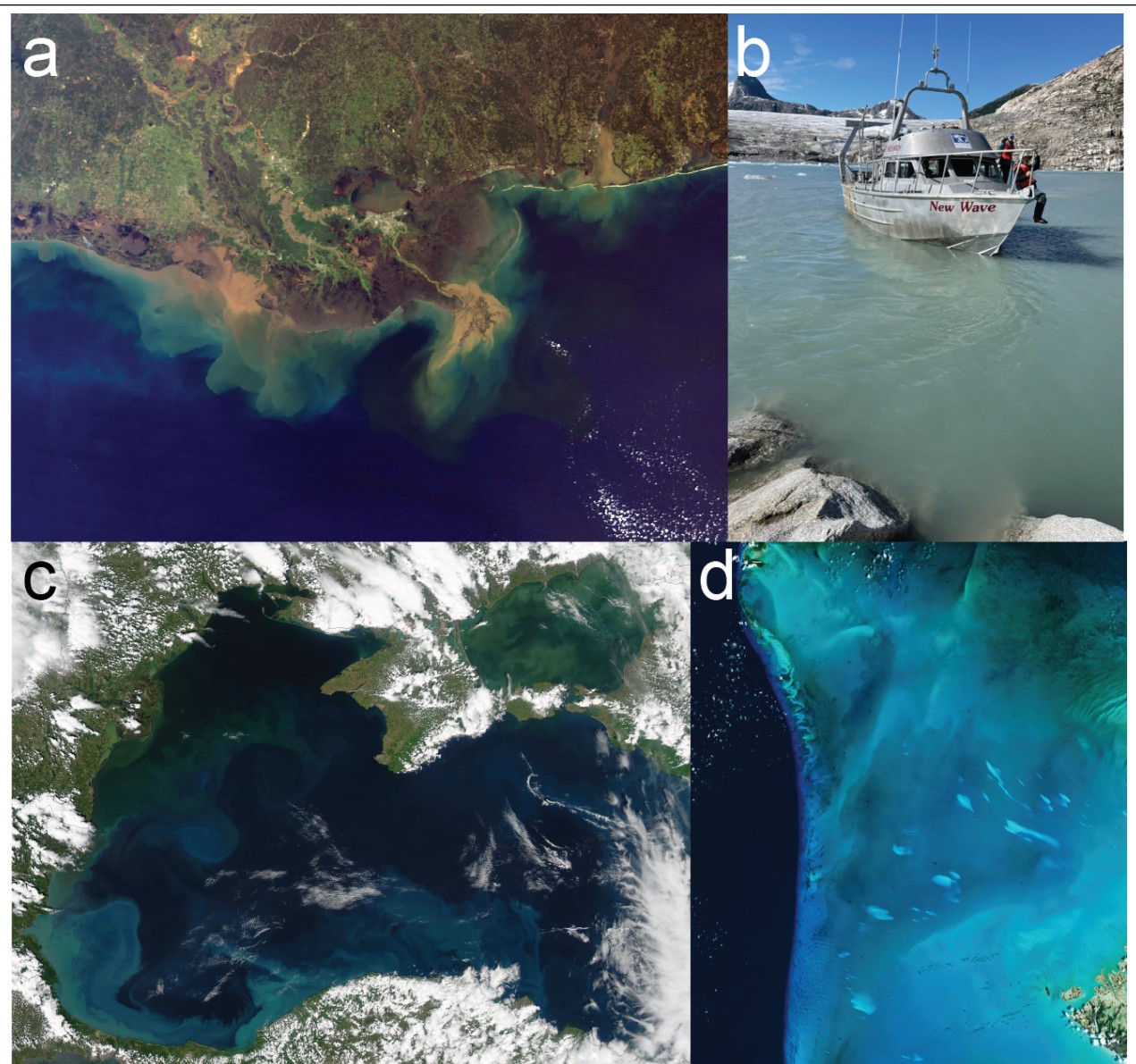

**Figure 1:** Images of some potential natural analog study sites. **a)** the Mississippi River plume in the Gulf of Mexico. **b)** a glacial fjord in Alaska, filled with mineral dust. **c)** Coccolithophore blooms in the Black Sea, visualized by satellite. **d)** Whiting events in the Bahamas. A, C, D are from NASA. B is from A. Gagnon.

that is delivered to the ocean via rivers. This input is balanced by global $CaCO_3$ burial in ocean sediments, along with significant contributions from other sedimentary processes such as groundwater discharge and nitrogen and sulfur cycling (Middelburg et al., 2020 and references therein). The $CaCO_3$ cycle buries roughly 36 Tmol alkalinity yr$^{-1}$ on shelves and along the coasts, and roughly 23 Tmol yr$^{-1}$ in the open ocean (Middelburg et al., 2020). However, open-ocean $CaCO_3$ production of

>100 Tmol yr[-1] greatly exceeds deep ocean burial, resulting in the recycling of ~77 Tmol yr[-1] via $CaCO_3$ dissolution to keep the system at steady state (Milliman et al., 1999, Berelson et al., 2007, Sulpis et al., 2021). Other mineral reactions, such as silicate weathering and reverse weathering, also produce and consume alkalinity within the ocean system. These alkalinity inputs, outputs, and internal cycles can serve as natural analogs to OAE, providing insight into how OAE deployments would interact with the ocean system, and what OAE deployments may look like at the gigaton scale.

Here, we define "Natural analogs" as Earth system processes that 1) resemble OAE deployments, or 2) can answer open questions about the feasibility, efficacy, and impacts of these deployments. Natural analogs can inform the deployment of OAE at a variety of scales, from small-scale field experiments to the global ocean. Natural analogs may offer test beds for sensor development across alkalinity and carbon gradients, could serve as real-world frameworks for interpreting laboratory and mesocosm experiment results, and could act as validation tools for modelers to study relevant OAE processes. In many cases natural carbonate chemistry parameters covary with other environmental variables such as temperature, salinity, nutrients, etc. Identifying alkalinity as the driver of a specific response in these systems can be challenging and must be carefully assessed. This drawback to natural analogs can also be a strength. Demonstrating the effect of alkalinity, in combination with a suite of other stressors or drivers, can be a powerful way to evaluate the downstream impact of OAE deployments, without the need for expensive and time-consuming field trials. Natural analogs, including periods of enhanced ocean alkalinity in the geological past, have the potential to elucidate longer-term, acclimated responses to OAE-relevant conditions.

## 1.2 The benefits and drawbacks of natural analogs

Natural analogs offer all of the benefits and drawbacks that come with the complexity of Earth systems. They should be viewed as one of many approaches available to OAE researchers. Manipulative experiments may be the most conclusive in terms of demonstrating immediate impact. Laboratory experiments (Iglesias-Rodriguez et al., 2023, this volume) offer ultimate control over conditions and variables, but their results can be challenging to apply to the real world. Mesocosms (Riebesell et al., 2023, this volume) are one step up in complexity, and benefit from not requiring field-trial permits to operate, but are costly and limited in their spatial and temporal applications. Field experiments (Albright et al., 2023, this volume) will provide the most information about real-world impacts. However, they require permits and resources that, currently, make them difficult and sometimes prohibitive to execute. In addition, none of these manipulative approaches can provide information on longer-term feedbacks or on large-scale processes. They may not last long enough to document adaptation of ecosystems to sustained alkalinity inputs. They also may be biased due to the timing and spatial limitations of these experiments, thus missing critical events such as the impact of weather, storms, turbidity flows, etc. Natural analogs can supplement these manipulative approaches in terms of complexity, scope, and scale. They will not necessarily give "clean" results for alkalinity effects alone; rather they offer a rich perspective on how OAE may look at scale.

Examples of recent studies of natural analogs in the context of OAE are still limited. However, previous research on ocean acidification (OA) highlights some of the difficulties and complexity associated with natural sites (e.g., Hall-Spencer et

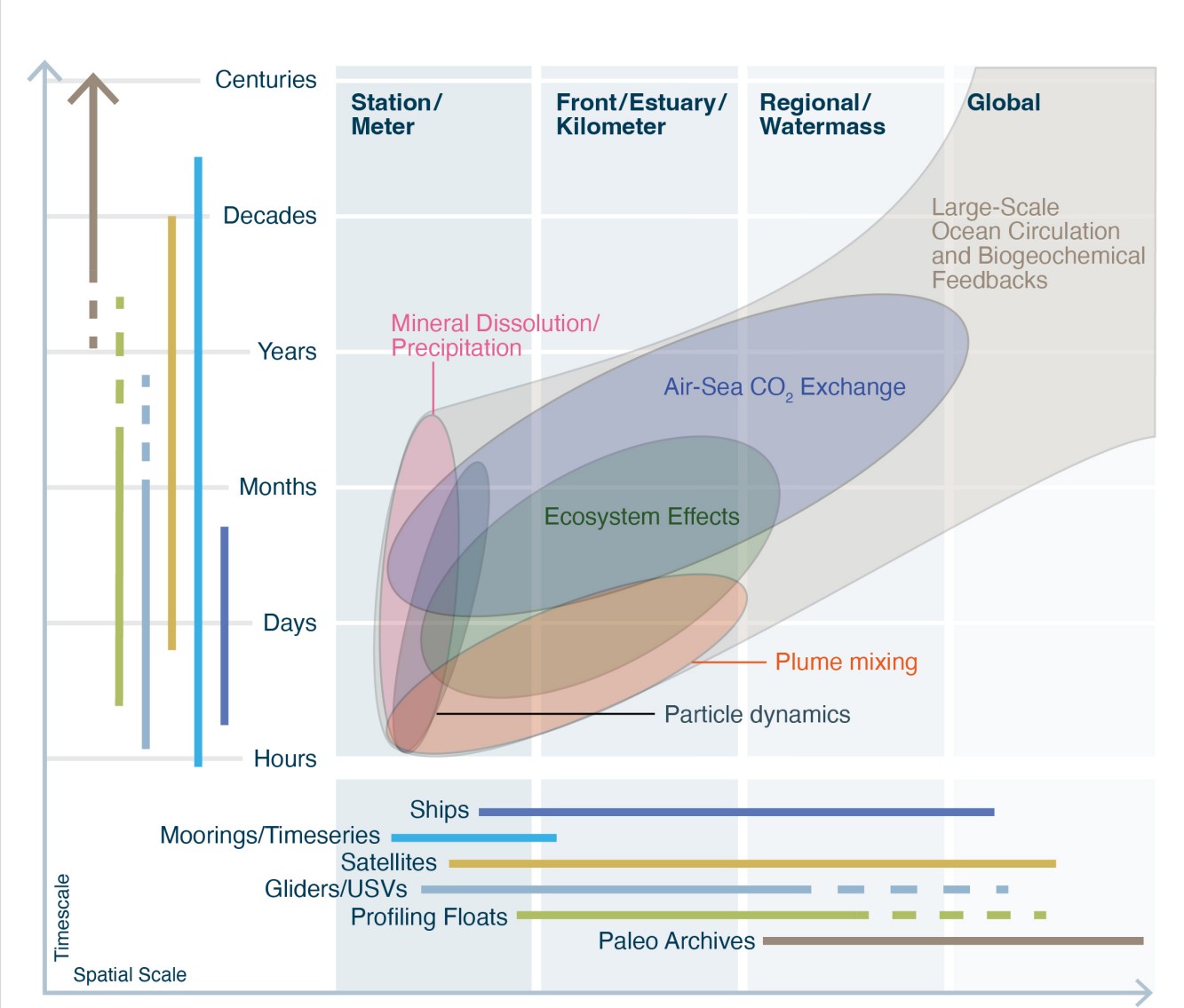

**Figure 2:** Processes relevant to natural analogs for OAE over a range of length and timescales. Various measurement platforms are shown in the margins, with their associated operating time and lengthscales. Studies investigating OAE should match measurement strategies to the appropriate processes being investigated. Figure concept adapted from Chai et al. (2020) and Bushinsky et al. (2019).

al., 2008, Tyrrell et al., 2008; Kroeker et al., 2013; Manzello et al., 2014; reviewed in Rastrick et al., 2018). Relevant examples for OAE may include large river plumes and estuarine systems where runoff into coastal systems - depending on catchment and underlying bedrock - may create interacting gradients in environmental parameters such as alkalinity, particulate matter,

dissolved inorganic carbon (DIC), salinity and/or macronutrients (Raymond & Cole, 2003; McGrath et al., 2016; Gomez et al., 2021), with each of these factors potentially triggering specific species-level or ecosystem responses.

Effects of covarying factors may be large and can wrongfully be attributed to the main variable or process of interest (in this case, alkalinity enhancement). To some degree, targeted site selection can minimize the number of confounding factors. Ideal locations for specific process studies would be sites with distinct spatial and/or temporal gradients in alkalinity and

limited fluctuations in other environmental variables (e.g., temperature and salinity, particulate matter, nutrients).

## 2 Some defining qualities of natural analogs

### 2.1 Earth system processes and their relationship to OAE

The delivery of alkalinity to the oceans via OAE will interact with the natural alkalinity cycle in various ways depending

on the approach, scale, and location of deployment. Accordingly, the relationship between OAE and Earth system processes will be expressed on a variety of spatial and temporal scales. We depict relevant Earth system processes as an oval, with its size and orientation determined by the temporal and spatial scale needed to characterize its influence in the Earth system (Figure 2). Mineral-fluid reactions, for example, can be studied in the lab at the (sub)micron scale of the mineral-seawater interface. This oval thus ranges from the bottom left corner vertically, to encompass a wide range of reaction rates at earth-

surface conditions. The effect of these reactions on seawater chemistry can occur across a wide range of scales, all of which require water mass transport. For example, at the platform or reef scale, observations must be made over days, weeks, or months to fully understand calcification budgets. Globally, the alkalinity of the ocean interior is increased through the dissolution of calcite and aragonite in the water column and sediments and the subsequent translation of that signal via ocean circulation.

Earth system processes do not operate in isolation, but instead overlap and interact with each other, creating higher-order effects that may generate unexpected and nonlinear responses at a range of spatial and temporal scales (depicted schematically as overlapping ovals, culminating in the gray envelope in Fig. 2). Reactions with minerals could ultimately engage with the carbon cycle and ocean-atmosphere $CO_2$ fluxes. Particle dynamics could feed back on mineral reaction rates, or begin to affect the biological pump, or both. Whether intentional field experiments engage these higher-order effects will depend on their

scale in both space and time. The benefit of natural analogs is that these effects are likely already fully coupled with each other. Studying natural analogs of OAE can thus test both (quasi) steady-state and transient effects associated with the interactions of these numerous Earth system processes. The large and at times undefined scale in both time and space presents a fundamental scale challenge for studying Earth system processes, and is a lesson that should be taken to heart by the OAE community. Below we discuss some of these Earth system processes that have direct relevance to OAE.

*Mineral precipitation and dissolution.* Silicate weathering is the most significant net carbon sink on geological timescales, and relevant dissolution reactions are occurring in many environments around the globe, including marine settings. These reactions are often slow, taking place on the timescale of months to years, or even longer. In the context of OAE, mineral

dissolution reactions will be limited to the treatment location where alkalinity production can be monitored. Because dissolution matrices are often complex (e.g. soils, sediments, and seawater), *in situ* dissolution rates are often hard to model and interpret. Interrogating real-world dissolution rates of these materials, either suspended in seawater or in sedimentary systems, would place useful constraints on dissolution rates and alkalinity production. Understanding real-world controls on secondary precipitation and subsequent alkalinity consumption will also be critical.

Carbonate minerals are considered some of the most reactive on the earth's surface, and their precipitation and dissolution occurs on faster timescales than most silicate mineral reaction rates. There is a major gap in our understanding of how OAE will interact with the ocean's $CaCO_3$ cycle. In the natural environment, biological and inorganic precipitation are related to a number of complex, interrelated factors. Surface seawater is already supersaturated with respect to most $CaCO_3$ minerals, and $CaCO_3$ precipitation is thought to be kinetically limited (Sun et al., 2015). Temperature, $Mg^{2+}$ and other ionic constituents, dissolved/particulate organic matter, and the *in situ* biological community, all may influence the rate and spatial extent of $CaCO_3$ precipitation. These factors will change both in space and time, meaning that the spatial scale of precipitation is often large and poorly defined.

*Particle dynamics.* Several OAE approaches involve adding fine-grained material to the ocean surface, and letting it dissolve to produce alkalinity. Currently, our understanding of how the sustained, large-scale addition of particles influence seawater turbidity, flocculation, particle settling velocities, and the marine ecosystem comes from the dredging and dumping literature (Essink, 1999). There may be additional feedbacks associated with mineral dissolution and precipitation reactions within the particle field. These particle dynamics occur on short spatial scales, but small particles could persist for long periods of time in the water column, leading to relatively long exchange timescales for some particle types and chemistries (Bacon and Anderson, 1982).

*Plume mixing and spreading.* The enhanced-alkalinity seawater plume resulting from an OAE deployment will be subject to a variety of physical forcings, and will spread out both horizontally and vertically over time. Plume dispersal will be influenced by currents, eddies, seabed topography, and other physical characteristics. Plumes of solid material will behave differently than plumes of dissolved alkalinity. The plume's dispersal will dilute its alkalinity, but will increase its surface area, creating tradeoffs for $CO_2$ uptake efficiency (He and Tyka, 2023, Wang et al., 2023). Alkalinity will also be lost below the mixed layer due to vertical mixing processes and circulation patterns.

*Ecosystem effects*. The ecosystem response to OAE is currently unknown (Bach et al., 2019). Responses may be quite variable, and will involve both immediate "shock" responses, and longer-term acclimated responses. Imagine exposing a marine ecosystem to a dispersing plume of alkalinity. Some parts of that ecosystem may sit directly in the outfall and experience sustained impacts, while others may experience periodic "whiffs" as the periphery of the plume disperses and shifts with water circulation. In the pelagic environment, the ecosystem may move along with the plume.

Whether alkalinity enhancement will stimulate biological calcification, either in open-ocean calcifiers such as coccolithophores, or in coastal ecosystems such as coral reefs or shellfish habitats, is an open question. In many cases, OAE will decrease the $pCO_2$ of seawater, potentially limiting the availability of $CO_2$ for photosynthesis for some organisms. In the case of solid additions, impurities and other constituents could dissolve along with alkalinity and could begin to interfere with the structure and function of marine ecosystems. How these effects are translated to higher trophic levels, and if there are any direct impacts on higher trophic level organisms, is poorly understood.

***Air-sea $CO_2$ exchange.*** The OAE approach to carbon dioxide removal (CDR) relies on the equilibration of an alkaline seawater parcel with the atmosphere. Air-sea gas exchange is thus a fundamental component of OAE and may play an important role in limiting the timescale of CDR. $CO_2$ dynamics and equilibration timescales are generally understood, and occur on timescales of several weeks to up to a year (Jones et al., 2014). The spatial scale of this equilibration requires the interaction of a water mass with the atmosphere, its physical and chemical characteristics, and wind speeds. It is well understood that increasing the ratio of total alkalinity to dissolved inorganic carbon (TA:DIC) in seawater decreases the partial pressure of $CO_2$, thus increasing the equilibration timescale of $CO_2$ uptake from the atmosphere. However, this process has not been investigated in practice. Natural analogs for OAE-induced $CO_2$ uptake could involve studying air-sea $CO_2$ fluxes at multiple locations with a range of surface seawater TA:DIC.

***Large-scale ocean circulation and biogeochemical feedbacks***. Eventually, and especially when considering OAE at gigaton scales, the processes listed above will blend with each other, leading to large-scale feedbacks of the biogeochemical ocean system. These feedbacks will become increasingly large and diffuse, essentially becoming part of the earth's biogeochemical cycling of alkalinity and carbon. Additionally, large-scale ocean circulation will redistribute alkalinity enhancements throughout the ocean interior. If OAE stimulates biological or inorganic $CaCO_3$ precipitation, alkalinity outputs could fundamentally change at the platform, basin, or global ocean scale. Ecosystem feedbacks, if sustained, could lead to significant reorganization of the biological pump with implications for the organic carbon cycle and the balance of $CO_2$ fluxes at the ocean surface.

## 2.2 A non-exhaustive list of OAE natural analogs

The processes and systems discussed in this chapter are not meant to be prescriptive or limiting. We encourage researchers to think creatively about the problems associated with OAE deployment – whether they be technical or scientific – and find suitable natural systems to study solutions to these problems. Many of the current open questions may get solved or become moot in subsequent years. The natural analog concept can, and should, continue to be applied even as our knowledge base for OAE grows and evolves over time.

Ideal natural analogs for all of the above processes, and how they will interact with OAE, will typically exist at system boundaries and across defined gradients in carbonate chemistry. For relevance to OAE, it will be important to constrain the

interactions between alkalinity and the system in question, and ultimately the associated implications for the efficiency, safety, and scalability of OAE.

***Rivers and their plumes and deltas (Fig. 1a).*** There may be opportunities to study natural river chemistries and their associated plume and sediment dynamics in regions with defined, sustained inputs to the marine system. Rivers deliver most of the alkalinity to the ocean, and dedicated surveys of these plumes across a variety of river compositions and plume geometries will provide critical information for large-scale alkalinity enhancement deployments. Deltaic environments may be useful to study the impact of particle loading and sediment-water interactions on the production and removal of

alkalinity (Wurgaft et al., 2021). We note that the TA:DIC of rivers is often very close to 1, such that alkalinity is often assumed to take the form of bicarbonate (Guo et al., 2012, Mu et al., 2023). Thus, riverine systems deliver DIC and TA in roughly equal amounts, limiting the utility of rivers as natural analogs for OAE processes with TA:DIC significantly different than unity. Alkalinity concentrations also vary between river systems due to the mineral composition of the drainage basin (e.g., high in the Mississippi and low in the Amazon), and as a function of discharge rates. These variations

can be used as comparisons or counterfactuals for natural analog studies.

    ***Glacial fjords and runoff into the marine system (Fig. 1b).*** The delivery, settling, and reaction of glacial flour in semi-enclosed or restricted basins could be useful for mineral dissolution/precipitation, particle dynamics, and plume evolution. Glaciers grind and dissolve underlying bedrock, creating fine-grained rock material known as glacial flour. This material

is often highly reactive, dissolving to produce cations and alkalinity in a wide range of concentrations (Brown, 2002). Mixing of freshwater and seawater has a unique impact on carbonate chemistry (Fransson et al., 2015, Horikawa et al., 2022), and could be linked to the source rock type and meltwater composition, among other factors. Glacial flour dissolution in seawater does not appear to be well-characterized, leaving a potential research avenue for OAE-related dissolution studies. There may be physiological impacts of glacial flour on the marine microbial community (Maselli et

al., 2023). Future studies on these glacial systems could inform deployments of fine-grained material for OAE and subsequent environmental monitoring strategies.

    ***Basin-scale systems with unique geochemistries and ecologies (Fig. 1c).*** The larger the spatial and temporal scale, the larger the natural analog system boundary must become. The advantage of natural analogs is that these large-scale

feedbacks can be assessed immediately. The Mediterranean Sea (Geyman et al., 2022), the Red Sea (Steiner et al. 2014), and the Black Sea (Bach et al., 2021) all provide unique high-alkalinity environments that could be compared and contrasted with more open-ocean settings to evaluate large-scale alkalinity and $CaCO_3$ cycling. Perhaps the most relevant question at this scale is evaluating whether basin-wide $CaCO_3$ formation is proportional to surface ocean alkalinity (or the ratio of TA:DIC). Basin-scale systems require large-scale observational capabilities such as repeat-hydrography cruises,

observational arrays, and satellite monitoring.

***The Bahamas carbonate platform and slope (Fig. 1d):*** Large plumes of suspended, fine-grained $CaCO_3$ appear regularly in the Bahamas and other marine locations, and their origin remains unclear. These events, known as "Whitings", have been studied since the 1930s (Black, 1933) and have been used to investigate the kinetics of $CaCO_3$ precipitation on calcite seeds (Morse et al., 2003). Studying the origin, duration, and extent of whitings would provide insight into how temperature and other seawater properties will interact with OAE to promote the formation of $CaCO_3$ minerals from seawater. Mineral precipitation could either occur directly from seawater (i.e. homogenously) or onto existing mineral seeds (i.e. heterogeneously). Studying whitings, and the mechanisms that drive them, may help elucidate the role that suspended sediments play in stimulating $CaCO_3$ precipitation (Broecker and Takahashi, 1966).

***Weathering of rocks on the seafloor.*** Since the discovery of hydrothermal vents, the alteration of rocks on the seafloor has been recognized as a major contributor to elemental mass balance in the ocean (Edmond et al., 1979). It is possible that seafloor weathering plays a significant role in controlling seawater calcium and magnesium concentrations, with implications for the marine alkalinity budget (Coogan and Dosso, 2022). Oceanic serpentinite is a common component of the seafloor, and contains significant portions of brucite that dissolve when exposed to seawater (Klein et al., 2020). Studying freshly drilled rock sequences could be used to study the precipitation and dissolution of Ca- and Mg- containing metastable minerals *in situ*. As a complement, studying the alkalinity balance of borehole fluid chemistry could help unpack how alkalinity is created and removed during seafloor weathering (Wheat et al., 2020), and help constrain the utility of basalts and other naturally occurring feedstocks for OAE.

***Phytoplankton blooms.*** During intense blooms, especially in enclosed or restricted systems, seawater pH can become measurably elevated relative to baseline conditions (Hansen, 2002). These events can be used to investigate potential ecological impacts of elevated pH (Pedersen and Hansen, 2003), as well as the potential for secondary $CaCO_3$ formation during phytoplankton blooms. These events are not directly caused by alkalinity enhancement, but DIC stripping, and thus may be more relevant as a natural analog to direct ocean capture of $CO_2$. Effects may also be hard to disentangle from nutrient/phytoplankton community dynamics due to vigorous biological activity.

***Beach locations with unique mineral sand compositions.*** Black sand or olivine beaches present unique opportunities to study integrated, long-term effects of mineral addition. There may also be opportunities to study ongoing beach nourishment projects that, while not strictly natural analogs, could provide systems for study without the need for additional permitting.

***Wastewater and other anthropogenic outfalls.** Although typically wastewater outfalls are acidic rather than basic, they represent opportunities to study the impacts of altered chemistries on the marine system. Again, these are not strictly "natural" analogs, but could provide useful information for the OAE research community.*

## 2.3 Extending OAE to geological timescales

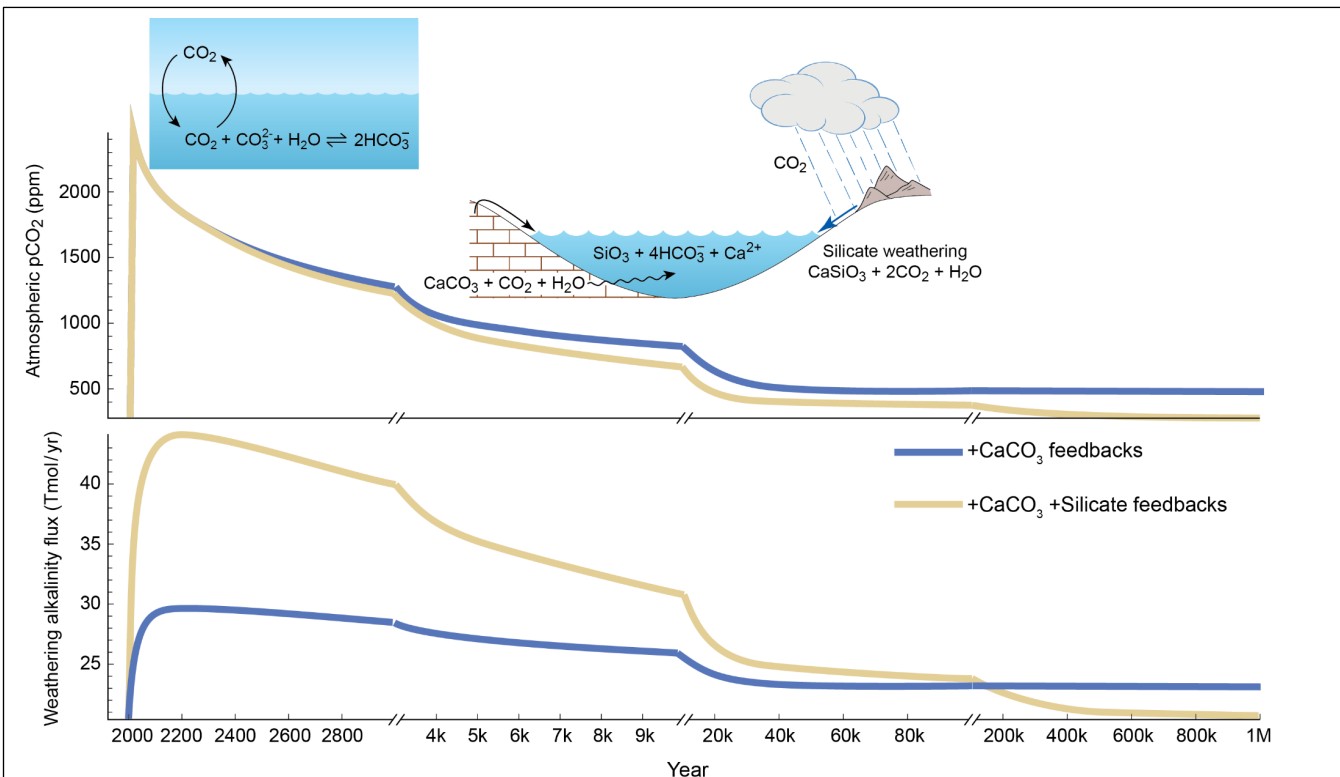

**Figure 3:** Evolution of (top) atmospheric pCO₂ and (bottom) weathering alkalinity flux to the ocean over 1 Myr for a 5000 Gt C emission pulse with terrestrial carbonate (blue) and silicate (tan) weathering feedbacks activated, using the GENIE Earth system model with a representation of terrestrial rock weathering. Note the changing timescale (adapted from Colbourn et al. (2015) where full details of the modelling experiment can be found). After the injection of carbon, pCO₂ initially declines very quickly due to invasion of the ocean and the onset of dissolution of seabed carbonates (a timescale of ~ 1000 years). The slower subsequent decline is due to the action of first carbonate weathering (up to ~ 10 kyrs) and then later silicate weathering feedbacks which reduce the pCO₂ to ~ preperturbation levels after ~500 kyrs when silicate weathering is active. The difference in the alkalinity inputs in response to the initial injection derives from the feedbacks associated with silicate weathering in GENIE from land temperature, run-off and productivity of the terrestrial biosphere.

The concept of ocean alkalinity enhancement as a means of carbon sequestration into the ocean is inspired by the conceptual mechanics of the long-term carbon cycle of the Earth system. Any additional output of acidic CO₂ to the ocean-

atmosphere system, derived from e.g. volcanic outgassing, is thought to be buffered naturally, and therefore stored in the ocean, on an expanding range of timescales by different components of Earth system alkalinity (Figure 3):

1. Dissolved carbonate alkalinity (titration of $CO_2$ with $CO_3^{2-}$ to yield $HCO_3^-$ redistributes additional carbon between the different carbonate species in solution (days-years)
2. Deep ocean $CaCO_3$ sediment (titration of $CO_2$ with seawater decreases the $CO_3^{2-}$ ion yielding lower carbonate saturation which drives deep ocean dissolution and release of alkalinity from carbonate sediments through vertical migration of the saturation horizon (7-10 kyrs)
3. Alkalinity released from increased weathering of silicate rocks as a result of elevated temperatures from additional $CO_2$ in the ocean-atmosphere system (Myrs).

***Geological measures of ocean alkalinity.***

Vertical migrations of the carbonate saturation horizon, at least since the advent of pelagic calcifiers ~220 Ma, moderate the deep ocean alkalinity burial to keep it in balance with the supply of alkalinity to the ocean from the release of cations through continental weathering (e.g. Broecker and Peng, 1987). Consequently, the carbonate compensation depth (CCD), defined as the depth beneath which there is no preserved carbonate in sediments, and which moves vertically largely in parallel with the saturation horizon, provides one of the best proxies for ocean alkalinity. Any deepening reflects increased ocean alkalinity and vice versa but not necessarily an increase in weathering inputs to the ocean.

A process of "biological carbonate compensation" can decouple the CCD from weathering due to environmental triggers which increase the shelf or pelagic carbonate production and burial above the CCD, and drive a shallowing or vice versa (Rickaby et al., 2010; Boudreau et al., 2019). Carbonate Ba/Ca and P/Ca have also been proposed as additional indirect measures of ocean alkalinity (Ingalls et al., 2020; Lea and Boyle, 1989).

***Geological targets to study ocean alkalinity enhancement.***

Over geological history, periods of elevated ocean alkalinity relative to carbon will cause an increase of deep ocean pH (traced with Boron (B) isotopes in foraminifera e.g. Foster et al., 2008) and/or carbonate saturation state. These periods are often defined relative to geological events, such as the Paleocene-Eocene Thermal Maximum (PETM), which corresponds to a very large injection of carbon into the Earth system resulting in deep sea ocean acidification and dissolution of deep sea carbonate. As a response to this pulse of seafloor alkalinity flux, the carbonate compensation depth overdeepened before restoring its equilibrium (Penman et al., 2016).

Geological periods of enhanced ocean alkalinity are characterised by either an increase in the source of alkalinity to the ocean, or a decrease in the sink. The major levers on the global alkalinity budget are those of weathering inputs and $CaCO_3$ burial but smaller contributors include reverse and submarine weathering and anaerobic processes.

Robust identification of enhanced weathering rates associated with e.g. elevated temperatures in the geological record could indicate a period of elevated ocean alkalinity due to enhanced alkalinity supply to the ocean. Disentangling weathering
intensity from isotopic proxies such as Sr, Os and Li isotopes is non trivial. Nonetheless, Earth's weathering thermostat does seem to be triggered to aid recovery after abrupt carbon perturbations when methane and/or $CO_2$ are added rapidly to the ocean-atmosphere system e.g. of the Mesozoic (Pogge von Strandman et al., 2013). Furthermore, towards the end of Snowball Earth events when volcanically sourced $CO_2$ builds in the atmosphere without a weathering sink due to the global ice cover, the post-Snowball Earth cap carbonates are taken as evidence of an abrupt increase of global weathering rates during the
hothouse aftermath of the Precambrian snowball events (Hoffman and Schrag, 2000).

Coupled deepening of the CCD with isotopic signals of weathering likely provide the best measure of events of ocean alkalinity enhancement. Due to the partitioning of carbonate sediments and alkalinity burial between the shelf and the deep ocean, any periods of lowered eustatic sealevel (such as sealevel regression, glacial maxima or ice house periods) which restrict the shelf area for carbonate burial, equate to elevated whole ocean alkalinity. This elevation occurs because the ocean
accumulates a greater alkalinity burden from weathering with a reduced shelf sink, potentially with enhanced carbonate alkalinity weathering from the exposed carbonate shelves, until the saturation horizon and CCD deepens. The aftermath of major extinctions involving extinction of biomineralisers, selectively or not, such as the Permo-Triassic, may be subject to enhanced ocean alkalinity in the aftermath as a result of the loss of a major biotic alkalinity sink (Payne et al., 2010; Knoll et al., 2007; Payne et al., 2007). Indeed immediately prior to the Cambrian explosion of skeletal organisms, both saturation state
and alkalinity are inferred to be highly elevated, from evidence of abiotic seafloor precipitation, due to the lack of a major biotic sink of carbonate (Grotzinger and Knoll, 1995).

Events of burial of organic carbon also perturb the TA:DIC budget by removal of DIC from the ocean-atmosphere system. Any reduction in DIC elevates the relative ocean alkalinity (and hence the TA:DIC ratio of the ocean and the deep sea carbonate ion) and can trigger deepening of the carbonate saturation horizon as seen during e.g. the regrowth of the terrestrial
biosphere at the end of the last glacial maximum (Berger, 1977). On the deglacial transition, there is a preservation spike in aragonite producing pteropods in the deep sea showing the elevated relative alkalinity in response to removal of carbon from the ocean-atmosphere system by biosphere regrowth.

## 3    Practical considerations for natural analog studies
The study of natural analogs is related to, but distinct from, basic research into the cycling of alkalinity and carbon through the Earth system. Because many different types of researchers may be approaching OAE and its interactions with the Earth system for the first time, we outline some practical considerations for field observations and the study of natural systems. There are both theoretical and practical constraints to conducting natural analog studies that should be taken into account when
determining the scope and scale of a campaign. Many of these concepts are either established in earth science, or in some cases, are still being actively developed as observational networks evolve and mature.

### 3.1 A primer on geochemical mass balance

The survey and sampling timescale is important when considering the spatial/temporal scale of your natural analog (Fig. 2),
the duration of the study, and the types of measurements and platforms used. When constructing a geochemical model of a
natural system, we typically make the assumption of steady state, or in other words that chemical concentrations are not
changing with time due to a balance between the inputs and the outputs on the timescale of interest. A steady state assumption

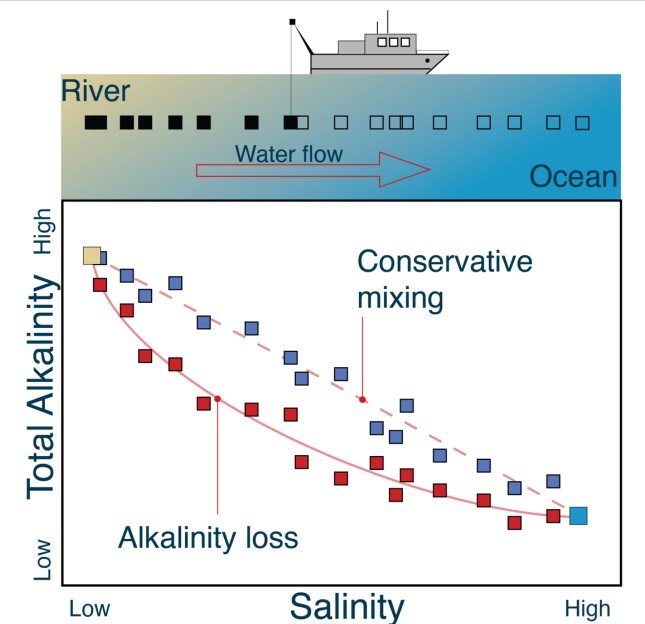

**Figure 4:** A schematic of a campaign sampling river water (left) out into ocean water (right). Salinity mixes conservatively, and by plotting total alkalinity (TA) versus salinity, researchers can examine and quantify processes that remove alkalinity from the system as the river flows into the ocean. Conservative mixing is illustrated by the blue points. Nonconservative mixing and alkalinity loss is illustrated by the red points.

allows researchers to assume that spatial gradients represent a balance of rates or fluxes – in other words, rates are now
expressed as a function of space rather than time. As an example, consider a natural analog study that is investigating the
removal of alkalinity as river water mixes with ocean water (Fig. 1a, Fig. 4). A survey is conducted, sampling down the river,
through the plume, and into the ocean. The river flow is unidirectional, and sets up a steady-state gradient between solutes in
the river and solutes in the ocean. The total salt content, measured as salinity ($S_{mix}$), is a proxy for the fractions of river and

ocean water in each sample, and an array can be constructed based on these measurements and the known salinity of the river ($S_{river}$) and ocean ($S_{ocean}$) endmembers (Boyle et al., 1974):


$$f_{river} = \frac{S_{mix} - S_{ocean}}{S_{river} - S_{ocean}}, \text{ (eq. 2a)}$$

where "$f_{river}$" is the fraction of river water in the sample. Assuming mixing with an ocean endmember, then the sum of the river and ocean fractions must sum to one:


$$f_{river} + f_{ocean} = 1 \text{ (eq. 2b)},$$

with "$f_{ocean}$" defined as the fraction of ocean water in the sample. As with salinity, TA and DIC are conservative quantities, meaning that they mix linearly and are not affected by temperature or pressure changes. If no chemical reactions are consuming
or producing TA or DIC along the flow path, then the samples will fall on a conservative mixing array, i.e.:

$$TA_{mix} = TA_{river} f_{river} + TA_{ocean} f_{ocean}, \text{ (eq. 2c)}$$

where the measured "mixture" value is a linear combination of the river and ocean endmembers. If the data fall on a straight
line between the endmember values for salinity and alkalinity, then alkalinity is not being produced or consumed in the system (e.g. the blue points in Fig 4). Any net removal or addition of alkalinity will move the data off of this conservative mixing line: Below it for removal, and above it for addition (e.g. the red points in Fig. 4). Critically, the signal of interest must be larger than the scatter in the data to quantitatively establish a reaction process, as illustrated by the scatter of data points in Fig. 4. At steady state, the alkalinity loss can be quantified by multiplying the river flux by the difference between these two curves.

The concepts of steady state and (non)conservative mixing are useful frameworks for setting up a study, interpreting the results, and quantifying biogeochemical processes over space and time, and can be applied to any water property that mixes linearly. In this river example, DIC-salinity relationships could be used for quantifying carbon uptake and loss due to gas exchange, photosynthesis, and $CaCO_3$ precipitation. While this method can diagnose net changes to TA or DIC budgets due to chemical reactions, the speciation of these quantities may change significantly along the flow path due to the nonlinear
nature of the carbonate system. For $pCO_2$ and pH, mixing relationships are more complicated because they are nonlinear functions of TA and DIC (Schulz et al., 2023, this volume). One important assumption in this model is that mixing only occurs between two endmembers. If there are more than two endmembers interacting in the study area, then this binary mixing model is not appropriate and deviations from linearity cannot be equated with chemical reaction. We discuss other methods of analyzing field data in Section 3.5.


### 3.2 Designing a suitable natural analog study

Determining whether a site is a suitable natural analog can be accomplished by asking a series of questions about its relevance to OAE deployment (Eisaman et al., 2023, this volume) and/or its monitoring, reporting and verification (MRV, Ho et al., 2023, this volume, Figure 4). When considering a candidate site, researchers should ask themselves: What qualities make the site relevant to OAE? Relevance can be clearly established through the presence of either enhanced alkalinity and/or solid materials that are producing alkalinity through interaction with seawater. However, other processes such as particle loading and plume mixing may be appropriate even in the absence of large alkalinity gradients (Figure 1).

The next consideration is timescale. How fast does the system change, and can alkalinity effects be assessed with an effective sampling strategy? Matching the measurement scheme to the process timescale (Figure 2, 3) is critical at this stage, and should guide the choice of measurement platform(s) (Section 3.3), and the associated measurement suite (Section 3.4). Alkalinity effects can only be assessed through to the counterfactual case, in a similar manner to how MRV will be conducted (Ho et al., 2023, this volume). A control or counterfactual, either in space or in time, should be established and should be quantifiable from the "OAE" condition (Section 3.5). The platform, measurements, and counterfactual conditions will all determine the approach for extracting alkalinity effects from the study location (Section 3.5). In some cases, it may be useful to pair field observations with models to contextualize your results (Section 3.6).

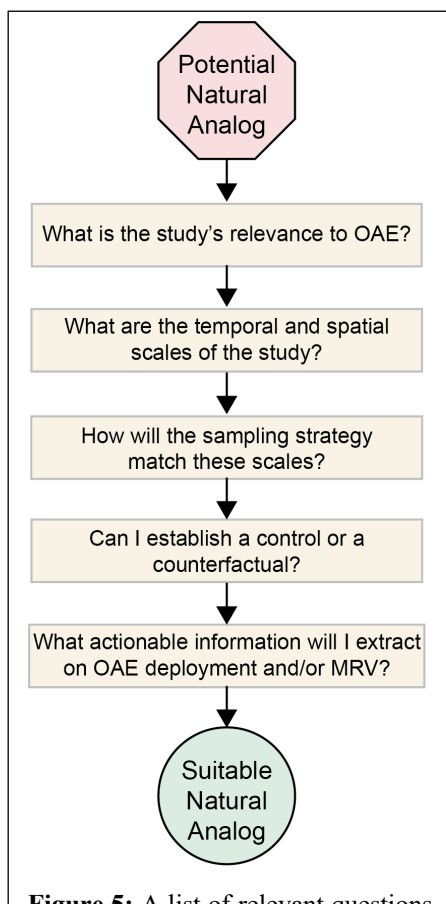

**Figure 5:** A list of relevant questions for deciding whether the study is suitable as a natural analog

In summary, given all of these considerations, it is critical to think through how the results of the study will be synthesized into actionable information about OAE deployment and its MRV. For example, if a feedback on alkalinity is established, can you relate that feedback to alkalinity loading to provide thresholds for OAE deployments, or minimum detection limits for MRV? Quantifying efficiencies on $CO_2$ uptake as a function of alkalinity loading are another example of a useful outcome from a natural analog. If these questions can be answered, then you have found yourself a natural analog. As shown in Fig. 1, rivers and their plumes, glacial fjords, the Black Sea, and whiting events in the Bahamas, immediately stand out as targets for natural analog studies.

### 3.3 Choice of platform

Once a potential process or site is chosen, it is important to consider what measurements are best suited for the study. Measurements can be conducted either in the lab or in the field, on vessels or remotely using autonomous assets (depicted in

the margins of Fig. 2). Choosing an appropriate platform and measurement suite will depend on the timescale of the process; access to equipment and instrumentation; and the practicality of the planned operations. Each platform operates within a specific window in both space and time, and these ranges should be considered when planning the field campaign.

***Research vessels*** have been part of the oceanographic toolkit for decades, from small boats all the way to 300+-foot global-class ships. These vessels offer flexibility, a range of built-in scientific instrumentation and equipment, research labs, and well-trained crew. Many research stations around the world operate their own smaller vessels that can be chartered for surveys. They can sample across entire ocean basins, but are limited in their temporal coverage to at most two months of continuous operation.

***Moorings and timeseries*** can be invaluable for studying a specific location over extended periods of time. Timeseries stations are critical for establishing the ranges of natural variability, and can be outfitted with a number of sensors and instruments. However, their applicability to a broader spatial scale is often limited, without other regional data or a model for context.

***Satellites*** can be tasked to investigate ocean-surface processes, and essentially cover the entire planet. Their timescales are often limited by their orbits, and thus cannot provide very high temporal resolution. Smaller constellations of cube-satellites can sometimes be tasked to give very high (sub-meter) resolution, and sometimes multiple transits in a single day, but their spatial scales are limited to coastal areas, and many are never tasked for open-ocean work. Satellites are capable of imaging the surface expression of bloom events (Neukermans et al., 2023) and $CaCO_3$
precipitation features such as whitings (Yao et al., 2023), as well as other optical properties such as chlorophyll fluorescence. They could potentially measure alkalinity if proxied by salinity (Priyanka et al., 2022). However, their utility for measuring water chemistry, especially below the surface, is limited.

        ***Gliders and Uncrewed Surface Vehicles (USVs)*** are becoming an important part of observational networks. Gliders
can provide high spatial resolution, but their operating speeds are often slower than crewed vessels. Depending on the operation, this limitation can be overcome by deploying glider fleets with a suite of intercalibrated sensors. Their operations are also limited by power, either batteries or access to solar or wind energy. Currently, gliders cannot sample the deep ocean, and can also not operate in very shallow or tightly constrained locations due to navigation constraints. Some vehicles can be piloted and reprogrammed on the fly, but many gliders have fixed trajectories that
are set upon deployment.

*Profiling floats* such as those used for the ARGO and Bio-ARGO program have been immensely helpful for establishing state estimates of the global ocean. Integration of biogeochemical sensors is ongoing. However, many profiling floats are limited to the open ocean, and cannot operate in coastal or shelf areas with shallow seabeds.


*Existing geological archives* such as sediment cores from drilling programs can be investigated for time periods and geological events that are relevant for OAE.

### 3.4 Choice of measurement suite

Oceanographic platforms host a unique set of measurements and capabilities, and come with tradeoffs between coverage, what you can measure, and how well you can measure it (Table 1, Bushinsky et al., 2019, Chai et al., 2020). Although bottle samples in the lab provide the highest precision and accuracy, they are limited in terms of sample throughput, preservation and shipping, and the need for expensive instrumentation. Underway or pumped systems can provide high-frequency surface data, but can clog/foul and need a source water to be pumped through them. They are best suited for research

vessels or moorings. *In situ* sensors can provide very high frequency data and be deployed on a range of platforms including gliders and profiling float. However, they must be calibrated, they can drift, and they are currently limited in what parameters they can measure. Remote sensing from satellites has by far the greatest spatial coverage, but is limited to the surface layer and by weather. Data are limited to optical measurements and imagery.

The natural variability of the site will be important to balance against your analytical capabilities. For instance, if an

estuary experiences tidal changes of >100 µmol kg$^{-1}$ alkalinity, it may not be informative to take daily samples without taking tidal cycles into account. *In situ* sensors with lower precision may not be able to detect small alkalinity enhancements above large natural variations. In addition, it is important to consider which carbonate chemistry variables are ideally suited for the sampling scheme. Alkalinity and DIC are both conservative and can be diagnosed with models as in Section 3.1, but in many cases pH may be more effective as a diagnostic tracer of multiple processes (e.g. alkalinity enhancement and subsequent $CO_2$

**Table 1:** Tradeoffs associated with various measurement approaches.

|      | Bottle Samples | Underway Systems | In situ Sensors | Remote Sensing |
|------|----------------|------------------|-----------------|----------------|
| Pros | • Very high precision + accuracy <br> • Well standardized <br> • All carbonate system parameters | • High frequency <br> • Immediate data <br> • Well standardized | • Very high frequency <br> • Can profile the water column <br> • Integration on a range of platforms <br> • Optical measurements possible | • Wide spatial coverage <br> • High spatial resolution |
| Cons | • Need for poisoning <br> • Requires laboratory <br> • Transport/shipping costs <br> • Lower temporal resolution | • Surface only <br> • Need vessel and pumped water flow <br> • Can foul/clog | • Limited carbonate chemistry parameter set <br> • Lower precision+accuracy <br> • Drift must be calibrated <br> • Integration is a challenge | • Limited temporal resolution (clouds, orbit) <br> • Must be ground-truthed <br> • Surrface only <br> • No carbonate chemistry, only optical measurements |

uptake). Combined with its relatively high measurement precision, frequency of measurement, and sensor availability, pH may be an attractive parameter for many early studies especially if it can be ground-truthed against alkalinity and DIC bottle data.

## 3.5 Establishing a control

In natural systems, there may not be a "perfect" control condition; instead, establishing relative changes between
conditions (spatial, temporal, etc.) may be all you can do. However, these relative changes should be clear and measurable given the sampling approach you have outlined. Controls can be established both in space and in time. For instance, different beaches, bays, or fjords can exhibit unique water chemistries and rock/sediment types. Setting up a similar survey or measurement scheme in two or more of these locations will yield a dataset that can be easily compared and contrasted.

Systems also change over time. For instance, the water chemistry, or river state, can be used to compare geochemical
processes when one endmember changes significantly from season to season. As an example, many rivers exhibit different solute concentrations and total water fluxes between the dry and rainy seasons. One season's survey can serve as a control for the second survey, provided that the conditions – and the expected geochemical signatures that result – change significantly on a seasonal basis. These conditions must be established in the context of the spatial and temporal timescales of the process of interest. In addition, similar assumptions for the steady state nature of the surveys should be verified, to ensure that the
results can be effectively compared.

## 3.6 Isolating alkalinity effects in your data

One of the main challenges when studying natural analogs in the context of OAE arises from potential concurrent effects of various confounding factors (e.g., temperature, salinity, nutrients, light, other carbonate system parameters) varying
in space and/or time along a gradient in TA. For example, alkalinity co-varies strongly with salinity on a global scale (Carter et al., 2014). Regional salinity-TA relationships may be better-suited for coastal applications, and may deviate significantly from this global relationship due to a number of biogeochemical processes (Hunt et al., 2021). Unequivocally attributing specific biogeochemical or ecosystem responses (e.g., $CaCO_3$ precipitation, species performance and distribution) to a single environmental variable (e.g., TA) remains challenging. Targeted monitoring combined with statistical tools can help to assess
the impact of confounding factors and identify relationships between various covarying factors and specific response variables. The choice of the statistical analysis depends on the particular question of interest and the complexity of the system to be studied.

*Multivariate analyses*, such as principal component analysis (PCA), are useful tools to determine the underlying
variability of a particular system without necessary predicting the relationship between a specific dependent and independent variable(s); e.g., to evaluate impacts of hydrography and carbonate chemistry on species performance (Kroeker et al., 2016) or to determine the main drivers of 'whiting' events (Yao et al., 2023).

***Simple and multiple linear regression (MLR) models*** are common tools to assess the relationship between a particular response variable and the variability in one or more predictor variables; e.g., to study links between changes in carbonate chemistry (e.g., TA, $pCO_2$, $\Omega_{Calcite}$) and biogeochemical or ecosystem responses (e.g., phytoplankton growth, calcification; Krumhardt et al., 2016; Silbiger et al., 2017). While general regression models come with clear benefits due to their simplicity, they can be restricting in their application given the assumption of linearity between dependent and independent variable(s). For example, DIC and TA mix linearly, but pH and $pCO_2$ do not. The model thus may perform poorly at capturing the complexity within certain data. In addition, regression models are highly sensitive to missing values and outliers, particularly in studies with a small sample size. It is advised to visually inspect the data and verify that the basic assumptions of the model are met before implementing a regression model. For example, graphical tools such as a scatterplot matrix and a bivariate correlation matrix help to verify that the relationships between dependent and independent variables are linear and independent variables are not highly correlated (e.g., no multicollinearity). Once a model has been implemented, additional useful validation tools may include histograms and Normal Q-Q plots to assess normality or scatterplots to check for constant variance of the residuals (i.e., homoscedasticity) across observations.

***Extensions to simple linear models*** may be applied in cases where particular assumptions are violated (e.g., non-linearity, non-normal distribution, heteroscedasticity). Possible modifications to the simple linear regression model include (1) generalized linear models (GLM) for non-normal distributions or (2) generalized additive models (GAM) for non-linear relationships. Machine learning approaches (such as neural network models and random forest regressions) are gaining increasing attention to address non-linearity, data complexity and data scarcity, and have proven skilful for generating predictive models to assess seasonal and inter-annual variability in carbonate chemistry across region and global scale (e.g., Bittig et al., 2018; Chen et al., 2019; Gregor & Gruber, 2021).

## 3.7 Regional modeling for field data validation

Studying natural analogs in the context of OAE has some clear limitations, largely due to the high complexity of the natural system and the difficulty in isolating the effects of TA from other environmental variables. Regional ocean models provide complementary tools that can help to disentangle the effects of confounding factors and determine underlying mechanisms driving observed patterns in the field. For example, Gomez et al. (2021) implemented a high-resolution ocean-biogeochemical model for the Gulf of Mexico to assess long-term trends in OA progression on a regional scale. By decomposing the carbonate system into individual components (e.g., $pCO_2$, pH, TA, $\Omega_{Ar}$), the authors showed that increased riverine alkalinity from the Mississippi River had a strong neutralizing effect on acidification near the river plume, thus may act as a key driver influencing the spatiotemporal variability of OA.

Regional models provide a verification framework for underlying physical and biogeochemical processes occurring in a system and, as such, can be a valuable tool to test our conceptual understanding of specific processes. Coupled physical-biogeochemical models to evaluate artificial ocean alkalinization on a regional scale are emerging (e.g., Butenschön et al.,

2021, Mongin et al., 2021; Wang et al., 2023; see Fennel et al., 2023, this volume, for details), yet similar modeling exercises applied to validate physical and biogeochemical processes along natural gradients are currently limited. In addition, many

models currently lack the ability to model the precipitation and dissolution of carbonate minerals either in these sediments or in the water column, especially metastable phases such as those found in reef environments. Implementing a regional model, for example, in areas where natural 'whiting' events occur (e.g., Bahama Banks) could be useful to test some of the various proposed mechanisms (e.g., abiotic/biotic calcification, sediment resuspension) leading to the observed accumulation of suspended calcium-rich particles in the water column (e.g., Larson & Mylroie, 2014; Yao et al., 2023). Recent model

simulation implemented a point-source OAE approach in the Bering Sea to evaluate the efficiency in $CO_2$ removal associated with a TA addition (Wang et al., 2023), but feedbacks associated with solid $CaCO_3$ cycling are currently missing from these modelling approaches. Similar approaches could give valuable insights when applied to natural analogs, for example to study the dispersal of an alkaline river plume and associated impacts on $pCO_2$ and carbonate chemistry, porewater alkalinity fluxes, or the interaction of mineral dissolution and circulation in enclosed basins.

In addition to hypothesis testing, models provide a means to increase the spatio-temporal resolution of *in situ* observations. The coverage of *in situ* observational data is often spatially and temporally limited due to logistical constraints (e.g., financial constraints, rare or remote location of natural analog) and/or natural variability of the system (e.g., seasonality, episodic occurrence), which can make replication challenging. Using ocean models in conjunction with natural (and field) studies allows to extrapolate spatially and temporally and fill in gaps in field observations.

In turn, models are evaluated in regard to how well observed patterns are reproduced, giving insights into underlying processes and how well these are represented in model parameterization. As such, model simulations rely on underlying assumptions that may not fully reproduce the high complexity and observational pattern of the natural system, in particular in regard to complex biological interactions (e.g., TA loss through carbonate mineral precipitation, trophic interactions, acclimation). For steady state systems (e.g., Black Sea), models do not explicitly resolve how the phytoplankton community

responds to chronic high-TA exposure. Natural analogs provide an opportunity to study long-term responses, and to continue developing modeling tools that are capable of resolving critically important biogeochemical processes.

### 3.8 considerations for future natural analogs

Studying natural analogs in the context of OAE is currently to some degree hindered by the availability and quality

of oceanographic data. The ongoing expansion of the observational infrastructure, including the deployment of autonomous vehicles such as gliders and BGC-Argo floats, continuously increases data coverage, quality and availability, making it progressively easier and cheaper to study natural analogs. *In situ* profiling platforms such as Argo floats are particularly useful for off-shelf regional and basin-scale studies. Autonomous platforms allow the expansion from remotely-sensed surface observations (e.g., satellite observations) to high resolution depth profiles, enabling the study of depth-resolved physical and

biogeochemical processes. Recent examples relevant for OAE include the depth-resolved detection of coccolithophores using

BGC-Argo floats (Terrats et al., 2020), increasing the spatial and temporal resolution of ship-based observations and expanding previous satellite-derived estimates to well below the surface layer.

**4 Conclusions and Key Recommendations**

The list of natural analogs and targets in the geological record highlighted in this chapter is by no means exclusive and additional suitable natural sites are likely to be identified as additional questions in the context of OAE arise. Key recommendations for the study of natural analogs include:

1. Appropriate consideration must be given to the spatial and temporal scales of the study, with implications for the scale of carbon storage potential represented at the site.

2. The measurement scheme and instrumental toolkit must be matched to the study site, in terms of scale, variability, and signal to noise. Models can be used to supplement *in situ* observations.

3. The study should be designed with applicable outcomes to OAE research, ideally with specific recommendations for deployment and/or monitoring of OAE at the study location or at places with similar Earth system processes.

Natural analogs with potentially different natural gradients, spatio-temporal resolutions and/or processes that are not accounted for yet in current surveys may be studied as sensor development and the ability to measure additional parameters evolve. Identifying key biogeochemical processes and ecosystem responses that can be measured and empirically linked to impacts of enhanced alkalinity is crucial in advancing our understanding of potential OAE impacts. Combining natural observational studies with controlled small-scale field manipulation or laboratory experiments will be key for addressing

knowledge gaps and questions in regard to specific biogeochemical reactions, spatial/temporal patterns and species interactions currently emerging from ongoing observational surveys. Importantly, no single approach will be able to resolve the full spatial and temporal extent and complexity of the system, and a combination of approaches (field studies, laboratory experiments, modeling exercises) will be required to address different physical and biogeochemical processes and levels of complexity.

**Author Contributions**

AVS, NL, and REMR conceived, wrote, and edited the paper.

**Competing interests**

The authors declare no competing interests.


**Acknowledgements**

This is a contribution to the "Guide for Best Practices on Ocean Alkalinity Enhancement Research". We thank our funders the ClimateWorks Foundation and the Prince Albert II of Monaco Foundation. AVS is supported by the Carbon to Sea Initiative, Thanks are also due to the Villefranche Oceanographic Laboratory for supporting the lead authors' meeting in January 2023.

AVS acknowledges support from the Carbon-to-Sea Initiative, a non-profit dedicated to evaluate Ocean Alkalinity Enhancement. We greatly appreciate helpful discussions and input from Alex Gagnon, and constructive reviews from Arvind Singh, Carolin Löscher, Andreas Oschlies, Jack Middelburg, and one anonymous reviewer.

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
