# Peer review of "Natural Analogs to Ocean Alkalinity Enhancement"

_State of the Planet, 2023_

## Author Comment (AC1)

**REVIEWER 1**

Review of Subhas, Lehmann and Rickaby

Enhancing ocean alkalinity is one of the potential carbon dioxide removal pathways needed to meet the Paris maximum global warming target of +1.5 and 2.0 ℃. To develop a verifiable carbon removal technique based on ocean alkalinity enhancement deployment, we must adopt a multifaceted research approach comprising laboratory studies, mesocosm, field experiments and natural analogues studies. This paper focuses on the latter and complements other contributions in this dedicated State of the Planet volume on Ocean Alkalinity enhancement deployments.

We thank Dr. Middelburg for their thoughtful review and constructive comments on the manuscript.

The paper is topical, timely and well-written, but it is not entirely clear what the target audience is. The text is for the major part quite generic and understandable for a wide audience, but sometimes too concise for a non-expert to grasp the message or too detailed given other parts of the text. As an example of the latter, the multivariate analysis is quite generic, and it is unclear why it should be discussed in two pages A4 given that OAE specific discussions might need some more elaboration (see below).

This point is well-taken, and we have adjusted the text according to the two reviewer comments. We have expanded some points while simplifying others in an attempt to keep the readership broad and inclusive. Specifically, we expanded the sections on describing relevant earth-system processes and potential natural analogs, making these components more specific and calling references where necessary. We streamlined the discussion of data analyses and added some sentences on machine-learning approaches.

Another major point of attention is that the manuscript needs some more articulation of the pros and cons of natural analogs vs. field experimentation/mesocosm studies. Field experiments and mesocosm studies do offer almost similar complexity while being more conclusive (manipulation experiments). However, these studies do not provide information on long-term feedbacks acting over larger space scale, neglect the adaptation of communities and organisms to enhanced alkalinity, and might be biased because of missing events (resuspension-deposition following storms, etc). The iron fertilization and ocean acidification literature could serve as inspiration.

We thank Dr. Middelburg for their suggestion of the OA and OIF literature. We have clarified the pros/cons associated with natural analogs for OAE studies in the

introduction accordingly, and made explicit the current limitations of small-scale field experiments.

Lines 21-25: These lines could be read as if alkalinity is primarily linked to carbon dioxide cycling and carbonate mineral dissolution and precipitation, and that other processes such as anaerobic mineralization are only locally of importance. There are multiple processes impacting alkalinity and for the presumed wide audience I would have expected at least something about the role of primary production/organic matter mineralization balance (after all we correct measured TA before we can use it to infer carbonate mineral dynamics) and that aerobic and anaerobic mineralization have different effects on alkalinity.

We have changed these sentences to make it clearer that the alkalinity cycle is intimately linked to ocean biogeochemistry, and has a number of influences including the cycling of organic matter.

Line 26:  The statement 60 Tmol alk y-1 needs a reference.

Thank you -- We revised this section and have updated references accordingly.

Line 27: I guess it should be Middelburg et al. 2020 not Middleburg et al. 2022.

Yes, apologies for the wrong year. This is now fixed.

Figure 2 and lines 92-109 (text on mineral dissolution and precipitation). The authors propose that mineral dissolution is occurring at smaller spatial scales than precipitation. Both can be studied at the micron-to-millimeter scales to regional/global scale, depending on the approach (experimental in the laboratory or computational chemistry for the small scale and budgeting for the ecosystem to global scale).  The presumed difference in spatial scale is unclear and needs further documentation. The alkalinity increase in ocean interior is primarily due to global scale calcite/aragonite dissolution.

The Reviewer makes a good point. We have now merged the "mineral dissolution" and "CaCO3 precipitation" into one process named "mineral dissolution/precipitation". Fig. 2 and the text has been updated accordingly.

River plumes are discussed as a potential natural analog. It would be instructive if the authors would communicate to the non-specialist audience that rivers often have similar DIC and alk concentration. Many freshwater chemists consider bicarbonate=DIC=alk. Rivers deliver alkalinity to the ocean, but in a 1 to 1 ratio with DIC.

A good point about the DIC-Alk balance in rivers. We have amended our discussion to include this point and the limitation of rivers to study DIC:TA ratios significantly different than unity.

Lines 110-115: There is an extensive literature on how dredging and dumping of mud can modify ecosystem functioning.

Thank you for this comment – we are not familiar with this body of literature, but did find a reference to point the reader to on this subject.

Line 140: Air-sea $CO_2$ exchange. One factor that might be mentioned here is the sensitivity of the seawater $CO_2$ to alkalinity addition. This can be explicitly calculated, and it might be useful to track in natural analogs.

This was not stated explicitly before, and it now is. Thank you.

Section 2.2. natural analogs.

When reading this list, I wondered why the Mediterranean or Red Sea with their natural high alkalinity are not mentioned. The combination Red Sea, Mediterranean, Black, Baltic and Atlantic ocean might perhaps be useful to tear apart the salinity-alkalinity co-variance.

We appreciate the Reviewer's insights here and add this to the "basin-scale" natural analog concept, along with a few relevant references.

Some other thoughts: alkaline seeps are less common than $CO_2$-rich seeps, but they do occur. Solar evaporation ponds with or without carbonate in the background might offer an excellent natural analog because at an evaporation stage of 2-2.5 times seawater one would expect a factor 2-3 difference in alkalinity between systems with or without aragonite (see seminal Laskar paper in L&O of early 80-ies).

We had a hard time finding this paper, but appreciate the suggestion!

Lines 221-242 The section on geological targets to study OAE is largely without references (or is there a reference limit to this publication outlet?).

We were given a strong guide to use few references with a rough reference limit. In light of this comment, we have added references to this section without over-burdening our reference list.

Lines 256-285 The conservative vs non-conservative estuarine mixing text needs some revision. Why is equation 1 presented explicitly? Is it really needed? The lines 280-285 could better articulate that (only) net removal or additions can be traced by non-conservative mixing.

We agree that Eq. 1 could be stated rather than written out, and have removed it. The point about net removal/addition is well taken and now included in the text.

For the non-specialist, it might be needed to explain here that DIC and alkalinity are conservative upon changes in mixing, T, P changes, while the species contribution to it, are not. (It is mentioned later, but not elaborated enough for a marine biologist, I guess).

A good point that has now been added.

Line 296: the abbreviation MRV appears without an introduction. Moreover, the Rau et al. paper could be useful to cite here.

Agreed, spelled out this abbreviation and cited as Eisaman et al., 2023, this volume and included in the references.

Line 305: Ho et al. is not in reference list

Thank you, this is now included in the references list.

Line 335: it would be useful to explain that satellites can track whitenings, cocco blooms, salinity etc, but alkalinity only if proxied by salinity.

Thank you, we added this point.

Line 366: perhaps reformulate or add:... may not be informative to take daily samples without taking tides into account.

Added, thank you.

Line 389: Alkalinity often co-varies with salinity, and this is probably the most challenging confounding factor to account for. Co-variance with temperature, nutrients etc will be far less. It would be useful to inform the reader about this well-known co-variance. (Salinity is often used as proxy for alkalinity in global mapping studies).

This point is well-taken and is now spelled out more clearly. We refer the reader to the alkalinity-salinity relationship of Carter et al (2014). and note regional deviations from this global relationship (e.g. Hunt et al, 2021).

Section 3.6 is rather long for its relevance for this paper. Yet machine learning approaches that better deal with non-linearities are not discussed.

Section 3.6 has been shortened and additional references for machine learning techniques have been added to the text.

Section 3.7. Many regional and global models resolving alkalinity dynamics lack sediment biogeochemical processes, process description for precipitation and dissolution and are limited to calcite. A few resolve aragonite as well, but high Mg calcite is not, which may be suboptimal when studying high alkalinity seas.

Agreed, this is an important point to make and we now include it in the modeling discussion.

Jack Middelburg, June 26, 2023

**REVIEWER 2**

Review of "Natural Analogs to Ocean Alkalinity Enhancement" by A. V. Subhas, N. Lehmann, and R. E. M. Rickaby

The authors provide an thorough overview of natural OAE analogs and how researchers could use these to study OAE effectiveness at ocean CO2 uptake, the impacts of OAE on ecosystems, mixing and spreading of alkalinity additions, and the interaction of these and other processes. They identify some examples of natural OAE analogs and discuss the benefits and challenges of using such systems to answer OAE questions. There are a variety of scales at which natural analogs can be found, from small scale river plumes or glacial fjords to large scale (and long timescale) processes such as changing carbonate compensation depth in the ocean. They provide an idealized example of a river with high alkalinity mixing with ocean water and explain how we can identify when more processes that affect alkalinity are at work than just mixing. The chapter further discusses how to determine if a location is a good candidate of a natural OAE study and proposes a framework for observing/measuring ocean variables and analyzing complex data to isolate the effects of changing alkalinity in a natural analog. The authors also touch on modeling efforts and highlight the utility of using models along with observations of natural systems.

This chapter is highly informative, well written, and I learned a lot reading it. It provides a high level summary of natural OAE processes and how we might use these to inform OAE research questions. Overall, I enjoyed reading the chapter, but I think there are few modifications the authors could make to make it even more enjoyable and understandable to a broad audience of researchers.

The biggest issue to me is that there a several places in the chapter where the authors touch on something that seems like it could be interesting and relevant but then don't provide enough background information for the reader to fully appreciate the concept. I realize that the chapter provides a high-level overview but sometimes mentioning something and then not providing enough detail can be a bit frustrating for the reader. Here are a few examples:

- Line ~175: What are whiting events and what do we know about them? Is the suspended CaCO3 coming from chemical precipitation or is the suspended CaCO3 delivered somehow? How long do they last and how frequently do they occur? Whiting events seem very intriguing as a OAE analog, but they need more of a background introduction. Perhaps a lot is unknown but that needs to stated explicitly.

We thank the Reviewer for prompting us to expand this section, as it was indeed rather brief. We have now expanded the discussion of whitings and included some relevant references. We also expanded the section on rivers and glacial fjords.

- Line ~180: What happens with seafloor weathering of basalts? Is there an example location of this? how does it change the water chemistry?

We have now expanded this paragraph as well and include some relevant references and locations.

- Line ~200: Figure 3 needs a more in-depth explanation to be useful to the reader. The 3-point list on lines 198 to 204 is helpful but perhaps reference the figure in this list. E.g., explain the difference between the tan and blue lines in each panel.

We have expanded the figure caption to explain the overall response with carbonate only and carbonate and silicate weathering feedbacks as described by the blue and tan lines respectively.

- Line ~209: It would be helpful to actually describe what happened during the PETM and how this affected dissolved alkalinity in the ocean. Just another sentence or two would be enough.

  We have described in greater detail the changes to ocean carbonate chemistry during the PETM associated with the initial acidification but then build up of alkalinity and overshoot in the CCD after the event and provided a reference for further details.

- Line ~235: What happened to biomineralizers during the Permo-Triassic ? Please describe more in detail.

  We have added some additional references to show the argument for severe acidification as a result of elevated CO2 at the PT boundary which contributed to the extinction, removed a significant biotic sink of calcium carbonate and lead to elevated alkalinity in the aftermath.

- Line ~450: the Gomez study needs to be more thoroughly described.. Please use a few sentences to say the purpose of their study and how the Mississippi river is bringing more alkalinity, and what they found.

  We have added a few more details about the Gomez et al. study, including the main goal and key finding, to the text.

All of the above points are interesting but need just a bit more explanation to be intriguing to the reader.

Minor comments:

Line 20: the authors say "at scale" several times in this section but it's not clear what this means. Do you mean at the scales meaningful for OAE deployments? Natural analogs vary in temporal and spatial scales so it's just not clear..

We thank the reviewer for this comment and now specify that, as you say, natural analogs can be used to address OAE deployments at a number of scales, from small-scale tests up to the global scale.

Line 28/29: define alk. it's obvious but should be defined before using it... or just write out alkalinity to be consistent with most of the rest of the chapter.. TA is also used quite a bit later in the chapter. Just be consistent with these abbreviations..

Thank you – we have written out alkalinity, and also define total alkalinity later on. Because total alkalinity is a measured quantity and has a technical definition, we leave its usage and abbreviation until later in the manuscript.

Line 58/59: run on sentence. Perhaps add a semicolon after "limited"

Thank you, we have made this change.

Figure 2: the x-axis needs a label (spatial coverage?)

We have added labels to the x and y axes (spatial coverage and timescale).

Line 81: Remove the word "always".

Done, thank you.

Line 87: Add "of OAE" after "natural analogs" just to remind the reader what you're talking about.

Done thank you.

Line 93: replace "should be" with "are"

Done, thank you.

Line 203: add "natural" before "environment"

Done, thank you.

Line 134: replace "are" with "is"

Done, thank you.

Line 140: what is meant by "surface expression"? the surface area?

Yes, we have reworded this section slightly to make the point that the water mass must interact with the atmosphere to exchange CO2.

Line 146: biological or chemical CaCO3 precipitation? or both?

Either or both – we have now clarified this point.

Lines 227 to 231: This sentence is really long and hard to follow. Please break it up and simplify or expand to make the subject matter understandable. So many subjects (e.g., snowball earth or carbon perturbations during the Mesozoic) are brought up that need more explanation to be meaningful.

Thank you, we have clarified the text accordingly and hope the Reviewer finds it easier to follow.

Line 242: Could you describe a bit more about the CCD deepening after the LGM with the regrowth of forests? Just needs a bit more to be intriguing to the reader who might not have a deep background in paleo research.

Thank you, we have clarified the text accordingly and hope the Reviewer finds it easier to follow.

Line 245: remove the word "here"

Done, thank you.

Line 256: concentrations of what? Alkalinity?

This section is meant to be general, and so we have specified "chemical concentrations".

There are two "Figure 4"s

Thank you, we have now changed the labeling of our figures.

The first Figure 4: Please define the red and blue squares in the figure caption. I realize that this is described in the text but it should also be in the caption.

Thank you, we have now added this description to the figure caption.

Line 265-275: Define "f_river" and "f_ocean"... at first I thought it meant fluxes but it makes more sense as fractions... so please describe better.

These are now defined in the text.

The second Figure 4: I found this flowchart a bit confusing.. I'm just wondering what the appropriate answers need to be to get to "Suitable Natural Analog"... So if the answer is "no" (or "none") for some of these questions, then it's not a suitable analog? To me, it just seems like a list of relevant questions to ask before selecting a OAE analog...

This is a fair point, and is even how we have phrased it in the text. We have redone this figure and changed the caption to "A list of relevant questions for deciding…"

Line 369: by "unmixed" do you mean differentiated?

We were trying to make the point that TA and DIC can be linearly mixed/unmixed as in Section 3.1, which we have tried to make clearer now in the text.

Line 429: I have no idea what homoscedasticity means.. perhaps explain in simpler language?

A simpler explanation of homoscedasticity has been added to the text.

Line 438: Reword using only the part in parentheses: "Given this higher flexibility, one of the drawbacks of GAMs is that non-linear features are potentially less intuitive and more complicated to interpret."

Thank you, we have changed the text accordingly.

Line 474: Is there a citation you could add for a reference that demonstrates that models cannot capture the adaptive response of phytoplankton to long term high-TA exposure?

We do not have a reference for this statement, but we are also not aware of a modeling study that explicity evaluates the response of phytoplankton to high-TA conditions. This is certainly an area for future study.

A final question for the authors: A variety of laboratory studies have shown that pelagic calcifiers, such as coccolithophores, calcify less as the ocean becomes more acidic. If natural coccolithophore populations start calcifying less due to ongoing ocean acidification, is this considered a natural OAE analog (since they start leaving more alkalinity in the surface waters since they calcify less)?

This is an interesting point, although it has not yet been shown that coccolithophores are calcifying less due to ocean acidification in the ocean (to our knowledge). We will leave this out for now, but it is worth considering in the future if it is shown to be the case!

---

## Author Response (AR2)

The authors have adequately responded to all of the reviewers comments and thereby considerably improved the manuscript.

We thank the handling editor for their positive feedback on the latest version of the manuscript.

I am satisfied with the revised version, except for the following points that require a minor revision:

p.3 first line: suggest to replace 'how OAE deployment will' by 'how OAE deployments would'

Thank you, we changed the wording accordingly.

line 70 small tests doesn't provide the reader with a clear picture of a scale. I suggest to replace test by something with a better define regional scale.

Thank you, we changed the wording to "small-scale field experiments".

line 86: I don't think it's generally correct that mesocosms are 'not requiring permits to operate'. I don't think you can deploy a mesocosm in most coastal waters without a permit. Suggest to rephrase, e.g. 'not requiring field-trial permits' or similar.

Thank you, we adjusted the wording accordingly.

line 176 typo: seawater

Thank you, fixed.

general: use consistent spelling of Earth system throughout the manuscript.

Thank you, fixed.

A section with "Key recommendations for researching natural analogs to OAE" (or similar section title with key recommendations in it) at the end is missing.
(See email by Angela of 23 August: Don't forget to include a "key recommendations" section at the end of your chapter. See p.63 of the OA Guide for a good example.)

We had included recommendations in the abstract, and have now included them explicitly in the conclusion as well.